# Post-Harvest Insect Pests and Their Management Practices for Major Food and Export Crops in East Africa: An Ethiopian Case Study

**DOI:** 10.3390/insects13111068

**Published:** 2022-11-18

**Authors:** Muez Berhe, Bhadriraju Subramanyam, Mekasha Chichaybelu, Girma Demissie, Fetien Abay, Jagger Harvey

**Affiliations:** 1Tigray Agricultural Research Institute, P.O. Box 62, Mekelle 5637, Ethiopia; 2Department of Grain Science and Industry, Kansas State University, Manhattan, KS 66506, USA; 3Debre-Zeyt Agricultural Research Center, Ethiopian Agricultural Research Institute, P.O. Box 2003, Addis Ababa 2003, Ethiopia; 4Bako Agricultural Research Center, Ethiopian Agricultural Research Institute, Addis Ababa 2003, Ethiopia; 5Department of Dryland Crop and Horticultural Science, Mekelle University, Mekelle 231, Ethiopia; 6Feed the Future Innovation Lab for Reduction of Post-Harvest Loss, Kansas State University, Manhattan, KS 66506, USA

**Keywords:** Ethiopia, stored commodities, post-harvest practices, storage pests, storage losses, pest management options

## Abstract

**Simple Summary:**

In Ethiopia’s main staple and export crops a significant number of storage pests have been found, and the losses caused by these pests were identified and reported. The effectiveness of post-harvest management options now available for Ethiopia’s staple and export crop pests was reviewed. However, based on the information that is currently available, it is challenging to rank the various storage pest management techniques. Cultural practices, the use of various locally accessible botanicals, varietal tolerance, various storage structures, and several significant entomopathogenic fungi are among the most frequently utilized management techniques. Although the majority of the research findings covered in this study are basic knowledge rather than ready-made technologies, current technologies can still be used with a few minor alterations. To regulate mold growth and lower the risk of grain storage pests, traditional subterranean pits can be slightly modified to avoid dampness. A further key factor in preventing field infestation and lowering insect loads in storage environments is the direct use of insect resistant varieties. Several botanical remedies have also been found to be successful, allowing farmers to use them with ease. Finally, it is critical to highlight the integrated pest management system’s sustainability and effectiveness of approaches for long-term storage of seeds and grains in a variety of situations.

**Abstract:**

Ethiopian subsistence farmers traditionally store their grain harvests, leaving them open to storage pests and fungi that can cause contamination of major staple crops. Applying the most effective strategy requires a precise understanding of the insect species, infestation rates, storage losses, and storage conditions in the various types of farmers’ grain stores. This study did a complete literature analysis on post-harvest pest and management measures with a focus on Ethiopia. The most frequent insect pests of stored cereals in this study were weevils (*Sitophilus* spp.), the lesser grain borer (*Rhyzopertha dominica*), rust-red flour beetle (*Tribolium* sp.), sawtoothed grain beetle (*Oryzaephilus* sp.), grain beetle (*Cryptolestes* spp.), Indian meal moth (*Plodia interpunctella*), and Angoumois grain moth (*Sitotroga cerealella*). Flour beetles (*Tribolium* spp.), sawtoothed beetles (*Oryzaephilus* sp.), flat grain beetles (*Cryptolestes pusillus*), and some moths have been identified as common stored product pests of stored oil seed, while bruchid beetles (*Callosobruchus chinensis*) and the moths were reported for pulses. Additionally, the storage pests in Ethiopia under varied conditions caused storage losses of 9–64.5%, 13–95%, 36.9–51.9%, and 2–94.7% in maize, sorghum, chickpeas, and sesame, respectively. To reduce the losses incurred, preventative measures can be taken before infestations or as soon as infestations are discovered. A variety of pest population monitoring systems for harvested products and retailers have been developed and recommended. In this context, reducing post-harvest grain losses is an urgent concern for improving food accessibility and availability for many smallholder farmers in Ethiopia and ensuring the nation’s long-term food security.

## 1. Introduction

By 2050, the world population has been estimated to reach 9.1 billion; this population requires a 70% increase in food production, which requires an increase in food supply by 60% to meet the food demand [1,2]. Globally, about 1.3 billion tons of food are wasted. In low-income countries, most grains and food are lost before reaching the consumer [1,2,3,4]. Different biotic (pests and disease) and abiotic (heat, drought, natural ripening process, and improper handling) factors are major constraints causing post-harvest losses and quality deterioration [1,5]. The losses that occur at various stages, such as in the field, storage, processing, and marketing in Africa, are frequently estimated to be between 20–40% [1,5]. These losses occur through various post-harvest activities, including harvesting, handling, storing, processing, packaging, transporting, and marketing. A significant food deficit is experienced in most “sub-Saharan” countries, and Ethiopia imports a significant amount of food every year to alleviate this food shortage [2]. Cereals, grains, legumes, and oilseeds make up the majority of Ethiopia’s sustainable agricultural production during the main crop season, which lasts from June to October. Some crops are also grown during the brief rains (March to May). Food security and supply in these circumstances relies heavily on appropriate storage practices. It is estimated that 60 to 90% of products are maintained in farm households and kept in storage for six to twelve months. [6]. Following harvest, this stock is vulnerable to losses and deterioration brought on by biotic and abiotic causes. Rodents, fungi, and insect pests are the main biotic agents that seriously disrupt storage. By eating grain, insects either directly or indirectly damage commodities. They can also contaminate commodities with their feces, webbing, and body parts [4].

Therefore, reducing post-harvest losses and deteriorating quality are the most crucial objectives for boosting food availability from established production, which might also improve food security and spur local economic growth. This paper offers a thorough literature assessment of post-harvest insect pests and their control strategies for important food and export crops with a focus on Ethiopia. Understanding post-harvest insect pests of important crops and focusing on intervention strategies to reduce the damage are the main goals of this effort. To achieve this goal, it is essential to that we study the different types of post-harvest insect pests that affect major Ethiopian crops, as well as the effectiveness and sufficiency of the available post-harvest management solutions across East Africa with a focus on Ethiopia.

## 2. Post-Harvest Insect Pests of Major Crops: A Worldwide Overview

Different researchers’ outputs have identified a number of post-harvest insect pests. Insects such as weevils (*Sitophilus* spp.), the lesser grain borer (*Rhyzopertha dominica*), rust-red flour beetle (*Tribolium castaneum*), sawtoothed grain beetle (*Oryzaephilus surinamensis*), flat grain beetle (*Cryptolestes pusillus*), Indian meal moth (*Plodia interpunctella*), and Angoumois grain moth (*Sitotroga cerealella*) are reported as pests of cereal grains, whereas flour beetles (*Tribolium* spp.), sawtoothed grain beetles (*O. surinamensis*), and moths have been reported in stored oil seeds [7]. For pulses, the bruchid beetle (*Callosobruchus chinensis*) was identified as a pest of stored products, among other bruchids [7].

There are two groups of stored grain insect pests, namely, “internal feeders” or “primary insect pests”, which develop and/or feed inside kernels, and “external feeders” and “secondary insect pests”, which develop outside of kernels and typically feed on damaged or broken grains [8]. Insect pests that are able to penetrate and damage the whole kernel of grain are categorized as primary insect pests. These pests include the rice weevil (*Sitophilus oryzae*), maize weevil (*Sitophilus zeamais*), granary weevil (*Sitophilus granarius*), lesser grain borer (*R. dominica*), pulse beetle (*Callosobruchus chinensis*), and Angoumois grain moth (*Sitotroga cerealella*). On the other hand, insects like the Khapra beetle (*Trogoderma granarium*), a devastating pest of stored grains, is an external feeder and is a quarantined pest. Other external feeders include the rust-red flour beetle (*Tribolium castaneum*), flat grain beetle (*C. pusillus*), sawtoothed grain beetle (*O. surinamensis*), grain mites (*Acarus siro*), and grain psocids *(Liposcelis divinatorius*), which cannot damage the whole grain but feed on broken kernels and damaged grain, are classified as secondary insect pests.

## 3. Importance of Post-Harvest Insect Pests on Major Food and Export Crops in Ethiopia

### 3.1. Maize

Pests impact maize production and productivity by attacking the roots, leaves, ears, tassels in the field, and grain during storage [9]. Important storage pests have been reported in maize stores, including *S. zeamais*, *S. cerealella*, the almond moth *Ephestia cautella*, *P. interpunctella*, the confused flour beetle *Tribolium confusum*, *Cryptolestes* spp., and sap beetles (*Carpophilus* spp.) [10,11,12]. Similar studies indicated that a number of stored pests, including the maize weevil, followed by grain moths, the rice weevil, and flour beetle, were among the important pests of stored maize in Jimma, Ethiopia. [11]. Most of the loss assessment studies carried out on maize indicated a significant loss of both grain and weight caused by different storage pests [10,12,13,14]. The majority of maize loss assessment studies showed that different storage pests cause significant grain and weight loss. According to this assessment, the total grain damage, ranging from 9 to 64.5%, and weight loss, ranging from 8.3 to 58.9%, were recorded under traditional farmers’ storage practices due to the maize weevil (*S. zeamais*), Angoumois grain moth (*S. cerealella*), rice weevil (*S. oryzae*), flour beetles (*Tribolium* spp.), and large grain borer (*P. truncatus*) (Table 1). This analysis also shows that the majority of loss assessment studies were conducted on *S. zeamais* damage, because *S. zeamais* is reported as the most important maize storage pest, causing significant quantitative and qualitative harm to traditional and improved maize varieties in Ethiopia (Table 1). A comparative assessment study on maize storage pests in different sites also revealed that *S. zeamais* was the most abundant and destructive storage pest recorded in all study sites; this pest accounted for 63.9% of the estimated grain weight loss during three to six months of maize storage [9]. Research works carried out on improved Quality Protein Maize (QPM) varieties also showed a loss of 37.8–53.1% in grains [13]. In general, the significant post-harvest maize loss sustained despite the poor overall agricultural output calls for an intervention to prevent the significant quantitative, qualitative, and economic losses caused by the storage pests during the periods when maize grain is stored.

### 3.2. Sorghum

The major storage pests in sorghum include *S. zeamais*, *S. cerealella*, *T. confusum*, and *O. surinamensis* in Africa, including Ethiopia [14]. Under various storage circumstances, like maize, this crop has also undergone serious damage, primarily by weevils, especially *Sitophilus* spp. [20]. Based on data collected through surveys and other standard methods, the total post-harvest loss ranges from 13–95% in different parts of Ethiopia under different farmers’ storage conditions (Table 1). Although most Ethiopian farmers in different regions of the country primarily store their grains in above-ground bins locally known as “gotera”, storing sorghum in traditional underground pits has also been noted as a common practice by most sorghum-growing farmers. This practice aims mainly to safeguard against burning, robbery, insect infestation, and domestic and wild animals [21]. In addition to storage insects, sorghum grains stored in underground pits were also extensively damaged by mold infections [2,4]. Significant sorghum grain and weight loss brought on by *Sitophilus* spp., *S. cerealella*, and *Tribolium* spp. were reported as a result of a substandard storage management method, and the damage observed worsened as the storage term was extended [20].

In general, insect and mold contamination in storage was blamed for the significant damage to sorghum-stored products, and under unfavorable storage conditions, it might even lead to a complete rejection of the grain (Table 1). To reduce the significant storage losses of sorghum grains, it is important to take into account various post-harvest loss management options as well as other elements in the sorghum production supply chain. The proper handling of the grain after harvest and before storage reduces the risk of storage infection and infestation and is a good action to take.

### 3.3. Chickpea

Chickpea productivity in Ethiopia is far below its potential due to different biotic and abiotic factors. Storage insect pests, including *C. chinensis*, the cowpea weevil *Callosobruchus maculatus*, *C. analis*, the bean weevil *Acanthoscelides obtectus*, and the bruchid *Bruchus incarnatus*, were reported to cause a significant loss in chickpea grain [16]. When large quantities of chickpea seeds are stored under poor conditions where they are sensitive to insect pest infestation, a severe infestation of chickpea post-harvest insect pests frequently results in a complete grain loss. [15]. In Ethiopia, subsistence farmers store their chickpea grain in conventional, insect-prone storage facilities for the majority of the time. As a result, insects from storage can attack the grains [16]. Severely damaged chickpea grains experience quality and quantity losses over the course of storage. According to an analysis of stored chickpea products in Ethiopia, *C. chinensis* damage resulted in a total weight loss ranging from 36.9 to 51.9% (Table 1). Additionally, a laboratory investigation revealed that *C. chinensis* induced a 50% weight loss on chickpea products that were kept for eight months in central Ethiopia [22]. In addition to the real loss, *C. chinensis*-damaged chickpea grain was also deemed unsuitable for food or feed, due to spoilage, foul odor, and toxin generation, as well as for planting due to poor germination [20]. Chickpea seeds suffer from direct physical losses and quality deterioration that influence the crop’s export value as well as its nutritional value, which directly affects the nation’s food security. Hence, conventional storage systems, along with improved storage technologies, can significantly reduce post-harvest loss damage during chickpea storage, which will help to mitigate the enormous losses incurred.

### 3.4. Sesame

Ethiopians refer to sesame (*Sesamum indicum*), often known as “Selit”, as a significant oil crop for export. However, it is frequently attacked by insect pests both before and after harvest all over the world. *Elasmolomus sordidus*, the sesame seed bug, is the most destructive post-harvest insect pest in Ethiopia. Some studies indicate that the sesame seed bug causes both quality and quantity damage to both sesame and groundnut seeds in the field as well as in warehouse conditions [17,19]. Due to the *E. sordidus* attacking throughout harvest, threshing, and storage time, a considerable loss in the weight of sesame, with a maximum of 94.7% and a minimum of 2%, was seen in Ethiopia (Table 1). In addition to the quantitative grain loss, these harmful pests also caused a 4 to 43% reduction in oil content and a 0.44 to 1.51% rise in free fatty acids [18]. According to an experiment carried out in Sudan, a maximum crop loss from sesame seed bugs was observed within 60 days of storage, and a significant reduction in oil content was also observed as a result of this pest damage [18]. Considering the increasing importance of this cash crop to the Ethiopian agricultural economy, alleviating the post-harvest management bottlenecks is key to reducing the damage caused by this important pest during pre-and post-harvest operational activities.

## 4. Post-Harvest Insect Management Practices in Ethiopia

### 4.1. Cultural Practice

While the enormous storage damage caused by a number of storage pests is successfully controlled by the use of improved storage technologies and various insecticides, most farms have also used a variety of conventional or cultural methods since time immemorial to manage pre- and post-harvest insect infestations. Improved varietals, good hygiene and sanitation, appropriate harvesting times, better storage facilities, and efficient drying are all part of this process. To keep their seed grain dry, Ethiopian farmers have stored their grain above the stove in the kitchen. When several cultural techniques were evaluated in Ethiopia, sun heating of maize resulted in a significant mortality (70–100%) of maize weevil under heating conditions of 55–60 °C for two–three hours utilizing solar heat absorption beds [23]. Similarly, the heat treatment of *C. chinensis* for about an hour in an obtuse-base-angle box heater lined with aluminum foil was also reported to completely kill adults of *C. chinensis* and also cause their failure to lay eggs in chickpea [23]. Although there are many different cultural practices used in various regions of Ethiopia, our assessment found that the most popular traditional approaches used by many farmers to reduce post-harvest loss include: drying grains to a safe moisture content before storage to prevent mold infection; avoiding mixing infected and healthy grains; heating at certain temperatures; adding inert powder to prevent damage by pests; cleaning before storage; smoking traditional storage with locally available plant materials; and other indigenous practices (Table 2). Clearly, there were differences in how well these generally accepted traditional methods controlled storage insect pest infestations and mold in both the field and storage settings. Therefore, based on the findings and suggestions from research, farmers and other participants along the value chain of a particular crop might employ a wide range of cultural strategies.

### 4.2. Storage Structure

In Ethiopia, traditional grain stores, such as “gotera” bags (made of polyethylene, sisal, or goatskin), “gumbi” earthen pots, and others, are the principal methods used to store grain. In a majority of the country, “gotera” (an above-ground bin) is the most widely utilized storage container. This above-ground bin is made of bamboo, which is plastered internally and externally with mud and cow dung and mostly placed outdoors [30]. “Gumbi”, on the other hand, is a tiered construction of rings placed one on top of the other and is made of mud, cow dung, and straw from crop leftovers such as teff and eragrostis tef [30]. Teff was also kept in various regions of the nation in traditional storage structures such as baskets, pots, gusgusha, barrels, and goggo [30]. According to research done in Southeast Ethiopia, 81% of the farmers stored their sorghum in “gotera”, whereas 17% used clay pots, and 1% used “gumbi” to preserve their sorghum after harvest [21]. Similar dried maize cobs were also kept in “gotera”, while farmers kept their harvested grain in polypropylene or jute bags after shelling and winnowing [31].

According to a comparison study of various storage structures, most farmers preferred to keep their cereals in bags within their homes (46%) and traditional “gotera” (39%) rather than use more modern storage such as metal silos (1%); this could be attributed to their lower cost and ease of access (Table 3). In Eastern Ethiopia, Harar, or cone-shaped subterranean pits with an average depth of 165.8 cm, a mouth diameter of 62.1 cm, and a bottom diameter of 152.0 cm, were also utilized for storing sorghum and corn (Figure 1) [21,31]. According to a survey performed in Harar, 70% of farmers kept their sorghum in the customary underground storage pits until they were used or sold [23]. Due to mycotoxin contamination, however, storing sorghum in subterranean pits proved less successful for preserving the quality [30]. Similar studies carried out in Ethiopia’s main food crop producing regions found traditional stores such as gotera (grain pits), bags (made of polyethylene, sisal, or goat leather), earthen pots, and others [32]. According to this study, more than 70% of the respondents stored their crop products in polyethylene bags and sacks, followed by the traditional gotera (67.8%), which is mostly preferred to store large quantities for a longer period of time (Table 3).

In conclusion, even though there are many storage technologies available for various grains, the choice of technology may depend on a variety of factors, such as the volume of production, the type of crop, the current weather conditions, the crop storage duration, the farmers’ ability and willingness to store the crop, as well as the cost-effectiveness of purchasing or implementing the storage structure for a given amount of crop produced. The post-harvest management strategy for a given crop under particular storage conditions could therefore be improved by extensive research to improve the capacity and efficiency of widely used traditional storage structures, as well as provide improved technologies to the community. This is due to the fact that these and other factors may potentially affect the capacity of safe storage.

In conclusion, although there are many storage technologies available for various grains, the choice of the specific technology depends on a variety of factors, such as the volume of production, the type of crop, the current weather conditions, the storage duration, the farmers’ ability and willingness to store the crop, as well as the cost-effectiveness of purchasing or using the storage structure for a given amount of crop produced. The post-harvest management strategy for a given crop under particular storage conditions could therefore be improved by extensive research to improve the capacity and efficiency of widely used traditional storage structures as well as provide improved technologies to the community.

### 4.3. Botanical Control

Plant products are seen to be effective and suitable for smallholder farmers to protect stored grain from insect damage. Treatment with the leaves of *Eucalyptus globulus*, *Schinese molle*, *Datura stramonium*, *Phytolacca dodecandra*, *Lycopersicum esculentum*, *Milletia ferruginea*, Mexican tea powder, triplex, filter cake, and neem seed were observed to cause high adult weevil mortality, reduced progeny emergence, and low grain damage of *S. zeamais* [27,29,34,35]. In a study on the management of the Adzuki bean beetle (*Callosobruchus chinensis*) using botanicals, inert materials, and edible oils in stored chickpeas, it was found that *Chenopodium ambrosioides* caused a high adult mortality, while the use of *Brassica juncea*, *Linum usitatissimum*, and *Guizotia abyssinica* seed oils caused a reduction in progeny emergence [34]. A study conducted to determine the effective concentrations of neem seed powder, citrus peel powder, and their oil extracts for effectiveness against maize weevil on sorghum varieties found that neem seed oil (NSO) and citrus seed oil (CSO) caused adult mortality in the range of 91.3–100% and a seed protection of 83–100% (Table 4). For the same botanicals, the study also revealed that weevil emergence, seed damage, and weight losses were statistically on par with the synthetic insecticide (Table 2). Similarly, beans treated with sun-dried powder of orange peel and an essential oil killed 65% and 67% of *Z. subfaciatus* after 96 h, respectively [26].

In general, this assessment revealed that insecticidal plant parts that are readily available locally play a crucial role in preventing pest damage in storage under various circumstances. In order to combat the significant infestation of storage pests, small-scale farmers might be encouraged to employ these readily available, affordable, and biodegradable plant products. It is critical to provide the target group with adequate knowledge on the formulation and application techniques, potential residual effects, toxicity to non-target species, and the ease of accessibility of those selected plant botanicals for practical use against target storage pests.

### 4.4. Use of Inert Dusts

Grain and seed storage have long been done using inert materials like wood ash, lime powder, sand, and other mineral dust. According to studies on the effectiveness of various insert dusts, using SilicoSec at 0.1% *w*/*w*, filter cake at 1% *w*/*w*, wood ash at 2.5 to 10% *w*/*w*, and sand at 30–70% was suggested as a substitute for reducing maize weevil damage under storage circumstances (Table 5). Similarly, coffee husk and wood ash at different rates also showed a good effect against the maize weevil [36]. An experiment done on grain wheat treated with ash and sand showed less grain and weight loss by *S. cerealella* and *Tribolium* spp. than the untreated grains [21]. A test conducted in the Gambella region also indicated that wood ash had a significant effect on managing bruchids on cowpea and a 90% reduction in F1 progeny of *C. chinensis* [37]. Similarly, cotton and Ethiopian mustard seed oils exhibit strong toxic activity against the Angoumois grain moth under laboratory experimental conditions [38]. In conclusion, the development of secure repellents against product pests is required due to the growing adverse effects linked with the usage of synthetic insecticides against stored product insects. Although different inert dusts, such as wood ash and other admixed grains, provide efficient protection against insect pests in storage, using inert dust in large amounts has several drawbacks. Alternative materials that could be effective at acceptable lower costs should therefore be given consideration.

### 4.5. Resistant Varieties

Different storage pest species react differently to different crop varieties for feeding and reproducing. Experiments conducted with hybrid maize varieties showed different levels of resistance to maize weevil and large grain borer [10]. In this case, maize genotypes such as AW8047, INT-A, Pob-62TLWF-QPM, TUXEPENO C6, USB, and Golden Valley were reported as comparatively resistant to the maize weevil, and these technologies can be used by resource-poor farmers [10]. Maize varieties with a tight and complete husk cover were selected by most farmers for its advantage of protecting against field infestation of the grain better than those with bare-tipped ears [24]. Similarly, significant differences in the storage resistance of haricot bean varieties to insect pests were also reported [39]. Likewise, 21 maize varieties were recently tested for resistance to maize weevil in the Bako Agricultural Research Center and the results of this study indicated that, based on the selection index, 6 were classified as resistant, 5 were rated as moderately resistant, and 8 were rated as moderately susceptible [33]. Currently, one weevil-resistant maize variety has been released by the National Maize Research Center [33]. In this approach, it is important to note that, for the effective use of resistant varieties against storage pests, the use of Marker Assisted Selection (MAS) methods can enhance the speed of resistant cultivar development. As a result, these new resistant types could be used as an affordable and environmentally responsible strategy to lessen post-harvest loss during storage. The resistant variants might also be a crucial part of an integrated pest control plan against pests that invade storage facilities.

### 4.6. Use of Entomopathogenic Fungi

Currently, many farmers and growers in developed countries are familiar with the use of predators and parasitoids for the biological control of arthropod (insect and mite) pests. However, it is also feasible to use specific microorganisms that kill arthropods. These include entomopathogenic fungi, nematodes, bacteria, and viruses. There are over 750 different types of fungi that can attack many insect and mite species simultaneously; although, some species and fungal strains have very specific targets [40,41]. Different experimental results in Ethiopia indicated that isolates of the two most common entomopathogenic fungi, *Beauveria bassiana* and *Metarhizium anisopliae*, showed a significant difference in mortality and the survival time for pests including the sesame seed bug and maize weevil (Table 5). The efficacy of 13 isolates of entomopathogenic fungi (*Beauveria*, *Metarhizium*, or *Paecilomyces* sp.) was assessed against *S. zeamais* and *P. truncatus* using a total immersion bioassay technique in the laboratory [28]. The result obtained indicated that all isolates tested were virulent to *P. truncatus* (98–100% mortality), while *M. anisopliae* and *B. bassiana* were virulent to *S. zeamais* (92–100% mortality); the isolate of *Paecilomyces* spp. was found to be the least virulent against *S. zeamais* (26.3–64.3% mortality). However, the pathogenicity and virulence level varied with the concentration and strain of the isolates [28]. According to this study, *P. truncatus* proved to be more susceptible to the entomopathogenic fungi tested than *S. zeamais* under Ethiopian conditions.

In general, there is a lot of pressure on farmers and growers to use fewer chemical pesticides. In order to combat storage pests, various control strategies must be sought out. The research to date suggests that utilizing entomopathogenic fungi may be a promising alternative strategy to manage the pests of stored products under particular storage conditions. Additionally, this technique could be used in Integrated Pest Management (IPM) programs.

### 4.7. Integrated Pest Management (IPM)

Integrated pest management emphasizes the integration of disciplines and control measures, such as varietal resistance, cultural methods, physical control, insecticidal plants, natural enemies, and pesticides, into a total management system to prevent pests from reaching damaging levels. However, only some reports on integrated management of post-harvest pests in Ethiopia have been available so far. In Ethiopia, the integrated use of the varieties, chenopodium plant powder, botanical triplex, SilicoSec, and filter cake against maize weevil was reported [34]. Generally, due to the costs and feasibility implications of using specific control methods, none of the various methods listed above can ensure safe storage. Therefore, it is crucial to combine all of the current pest control techniques, including biological control, cultural approaches, resistant genotypes, and other non-polluting techniques, in order to develop a post-harvest loss management strategy that is both affordable and long-term.

## 5. Conclusion

To increase food security, particularly in low-income nations like Ethiopia, the issue of post-harvest losses is the top item on the agenda. For Ethiopia’s main food and export crops, a number of significant storage pests have been found, and the losses caused by these pests have been identified and reported. An assessment was made as to how well the current post-harvest control solutions worked for the main pests of Ethiopia’s food and export crops. However, based on the information that is currently available, it is challenging to rank the various storage pest management techniques. Cultural techniques, the use of various locally accessible botanicals, varietal tolerance, various storage structures, and several significant entomopathogenic fungi are among the most frequently utilized strategies discussed here. Although most of the research findings covered in this study consist of fundamental knowledge rather than ready-made technologies, the current technologies can still be used with a few minor alterations. The classic subterranean pits, for instance, can be slightly modified to prevent dampness, thereby preventing the spread of mold and reducing the risk of storing pests. A further key factor in preventing field infestation and lowering insect loads in storage environments is the direct use of the specified resistant types. Several herbal remedies have also been successful, allowing farmers to use them right away. Last but not least, it is critical to highlight the integrated pest management system’s viability and effectiveness as a strategy for long-term storage of grains and seeds under a variety of circumstances.

## Figures and Tables

**Figure 1 insects-13-01068-f001:**
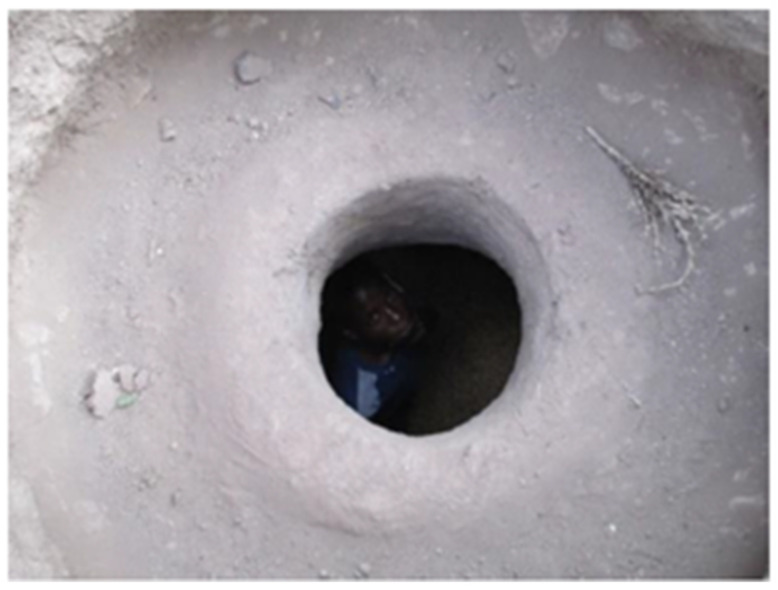
Large traditional underground pit for sorghum in Fedis, Ethiopia. Source; taken from FAO, 2017 [32].

**Table 1 insects-13-01068-t001:** A comparison of storage losses and pests of Ethiopia’s main food and export crops.

Crop	Cause of Loss	Estimated Grain Loss %	Estimated Weight Loss %	Reference
Maize	*S. zeamais*, *S. cerealella*, *S oryzae*, *T. confusum*	64.50%	58.85%	[12]
	*S. zeamais*	NA	63.85%	[13]
	*S. zeamais*	12–20%	NA	[10]
	*P. truncatus*	9–45%	NA	
	*S. zeamais*	25–46%	NA	[13]
	*S. zeamais*	18.0 ± 3.4%	8.3 ± 0.2%	[13]
	Storage insect	37%	NA	[2]
	Storage insect	13%	NA	[2]
	Insects & molds	90–95%	NA	[4]
Sorghum	Insect	50%	NA	[14]
Chickpea	*C. chinensis*	36.9–51.9%	NA	[15]
	*C. chinensis*	50%	NA	[16]
Sesame	*Elasmolomus sordidus*	2–36%	25%	[17]
		13%	26%	[18]
		94.7%	23%	[19]

NA = data not available; Source (analyzed by the author).

**Table 2 insects-13-01068-t002:** Comparison analysis on effectiveness of storage pest control methods in Ethiopia.

Control Methods	Crop/Trait	Treatment	Target Pest	Measurement Scale	Efficiency	Reference
Cultural	Maize	Solar heating (55–60 °C)	*S. zeamais*	Mortality rate %	70–100%	[23]
Variety screening	Maize (husk level)	G1	*S. zeamais*	NWPEAM	59.15 ± 8.13 a	[24]
		G2			12.78 ± 1.44 c	
		G3			15.85 ± 7.76 c	
		G4			32.45 ± 8.09 b	
		G5			10.55 ± 1.94 c	
	Maize (Biochemical level)	Pratap makka-5	*S. cerealella*	Grain loss %	7.21%	[25]
		PMH-1	*S. cerealella*		31.12%	
Botanicals	Haricot beans	Orange powder (96 h)	*Zabrotes subfasciatus*	Mortality rate %	65.9%	[26]
		Orange essential oil (hours)		Mortality rate %	67.4%	
	Sorghum	Neem seed oil	*S. zeamais*	Mortality rate %	91.25–100%	[24]
		Citrus seed oil			83–100%	
	Maize	Cooking oils	*S. zeamais*	Mortality rate %	54.54%	[27]
		Triplex			92.66%	
		Chemicals			100%	
		control			24.2%	
Entomopathogenic fungi	Lab	*Metarhizium anisopliae* & *B. bassiana*	*S. zeamais*	Mortality rate %	92–100%	[28]
		13 isolates of	*P. truncatus*		98–100%	
Inert dust	Maize	SilicoSec rates (15 days)	*S. zeamais*	Mortality rate %	100%	[29]
		Filter cake rates			100%	
		Wood ash rates			98.7%	

(NWPEAM = number of adult weevils per ear after a month of storage); Source (analyzed by the author). Means in columns with the same letter are not significantly different.

**Table 3 insects-13-01068-t003:** Post-harvest storage methods in Ethiopia expressed as % of the total number of storage methods used for each cereal type.

Survey No.	Type of Structure	Respondents (%)	Grains Stored (%)	References
1	Bags	70.5	46	[2,32]
2	Gotera	67.8	39	[2,32]
3	Pots	9.4	NA	[33]
4	Underground pits	0.3	NA	[32]
5	Metallic silo	NA	<1	[2]
6	Others	19.1	14	[2,32]

NA = data not available; Source (analyzed by the author).

**Table 4 insects-13-01068-t004:** Effect of NSO and CSO on weight loss, damage, and germination.

Treatment	Rate (mL or g/0.1 kg)	Weight Loss %	Damage %	Germination %
CSO	1.50 mL	0.3 f	1.1 f	70.3 bc
	1.00 mL	1.6 e	4.7 e	73.3 b
	0.75 mL	1.9 e	5.6 e	62.5 bc
	0.50 mL	3.1	8.5 d	64.8 bc
	0.25 mL	5.9 c	14.6 c	65.8 bc
NSO	1.50 mL	0.0 f	0.0 g	59.8 bc
	1.00 mL	0.0 f	0.0 g	73.8 b
	0.75 mL	0.3 f	0.9 f	77.0 b
	0.50 mL	7.6 b	20.6 b	55.8 bc
	0.25 mL	9.6 a	25.5 b	56.0 cd
Malathion5% dust	0.05 g	0.0 f	0.0 g	92.3 a
Acetone treated	2.00 mL	10.7 a	31.9 a	47.8 c

Means in columns with the same letter are not significantly different at α = 0.01. Mean separation was analyzed by the Student–Newman–Keuls test; CSO = Citrus Seed Oil; NSO = Neem Seed Oil; Source; Kifle et al. [26].

**Table 5 insects-13-01068-t005:** Main effects (±SE) of inert dust (SilicoSec, filter cake, and wood ash) and their rates on the percent of the mortality of adult maize weevil 3, 7, and 15 days after exposure.

Main Effect	Inert Dust/Rate (% *w*/*w* of Grain)								
	SilicoSec Rate			Filter Cake Rate			Wood Ash Rates		
	0.05	0.1	0.2	1.0	2.5	5.0	2.5	5.0	10
3-day after exposure									
	98.8 ± 1.3 a	99.1 ± 0.4 a	99.5 ± 0.9 a	87.4 ± 2.4 cd	92.3 ± 7.8 bc	97.8 ± 7.5 ab	65.0 ± 6.9 f	73.2 ± 12.7 ef	79.6 ± 5.5 de
Inert dust		99.1 ± 0.43 a			92.5 ± 1.58 b			72.6 ± 3.29 c	
7-day after exposure									
	99.4 a	100 a	100 a	95.8 ± 0.67 ab	98.5 ± 1.33 ab	99.6 ± 4.22 a	84.7 ± 6.57 c	89.0 ± 8.04 c	93.7 ± 5.22 c
Inert dust		99.8 a			97.9 ± 0.64 b			89.1 ± 1.61 c	
15-day after exposure									
	100 a	-	-	100 a	100 a	100 a	97.6 ± 1.11 b	99.3 ± 2.0 ab	99.3 ± 0.67 ab
Inert dust		100 a			100 a			98.7 ± 0.37 b	

Means with the same letter in a row are not statistically significant at α = 0.05. “-” Data not available as all treated insects died. Source: Girma et al. [13].

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
