# Peer review of "Post-Harvest Insect Pests and Their Management Practices for Major Food and Export Crops in East Africa: An Ethiopian Case Study"

_insects, 2022, doi:10.3390/insects13111068_

Round 1

Reviewer 1 Report

I recommend this manuscript for publication after the authors make the necessary changes listed in the reviewed version in the attachment.

Author Response

We appreciate that the reviewer’s comments. The followings are our point-by-point responses:

  1. Spelling errors

 Responses:  we have corrected all the spelling and grammatical errors  suggested

  • Pets corrected to pests : line 82, 105
  • Milletia ferruginea corrected to Millettia ferruginea: line 289
  • Oryzaephillis corrected to Oryzaephilus surinamensis: line 144
  • Milletia ferruginea corrected Millettia ferruginea: line 289
  • Cussed corrected to caused: line 295
  • Beans corrected to beans: line 302
  • rust- red flour beetle corrected to red flour beetle : Line 105
  1. Put scientific and journal names in italic font

Responses: we put all insects’ scientific name and journal name in references in italic font

  • zeamais formatted to S. zeamais: line 103, Table 1, Table 2 (revised version)
  • Sitotroga cerealella formatted to Sitotroga cerealella: Table 2 (revised version)
  • Prostephanus truncates formatted to Prostephanus truncates: Table 2 (revised version)
  • large grain borer formatted to Prostephanus truncates : Table 1
  • Elasmolomuse sordiduse corrected to Elasmolomus sordidus : line 192, 196
  • Schinese molle corrected to Schinus molle: line 289
  • subfaciatus formatted to Z. subfaciatus : line 303
  • Metharizium anisophilae corrected to Metarhizium anisopliae: line 368
  • Beauveria, Metarhizium, Paecilomyces spp., Sitophilus zeamais, Prostephanus truncates, B. bassiana, S. zeamais, and truncates,     formatted to Beauveria, Metarhizium, Paecilomyces spp., Sitophilus zeamais, Prostephanus truncates, Beauveria bassiana, S. zeamais, and  P. truncates    and  respectively: line 368 – 378
  • International Journal of Sciences: Basic and Applied Research formatted to International Journal of Sciences: Basic and Applied Research: line 464
  • Archives of Biological Sciences formatted to Archives of Biological Sciences formatted.
  1. Other specific points
  • We formatted citations according insects format; [1,2,3,4] formatted [1-4]
  • In the introductory part, we switched from bullet points to text for our objects.
  • Revisions to the manuscript are marked by “Track changes” function

Reviewer 2 Report

Dear Authors,

The manuscript is interesting but difficult to understand. Did the last author review and edited prior to submission? You need to further improve the language and meticulously correct the latin names and their authorities. Format needs careful attention since in numerous points (and the references) is not in aggrement with the format proposed by Insects. Please see my detailed review in the attached annotated pdf file.

Author Response

We appreciate that the reviewer’s comments. The followings are our point-by-point responses:

  1. You need to further improve the language and meticulously correct the latin names and their authorities.

            Response:

  • Language and grammar have undergone considerable revisions in response to insightful reviewer’ suggestion.
  • All Latin names, incorrectly spelt words, and poorly formatted grammar have been corrected.
  • accurate information about the authors’ and the institutes’ provided in line with insects format
  1. please choose ‘various or many’

              Response:

  • we have deleted ‘many’: line 14
  1. What do you mean by major food? Please explain

              Response:

  • the word major food is changed to staple crops which means crops grown in large for food: line 15
  1. ‘discusses’ what do you mean? Is it written correctly? line 19

              Response:

  • we corrected to ‘management methods discussed’:
  1. Please remove examples from the simple summary: line 23

              Response:

  • we removed
  1. Please use unified format for ‘9%-64.5%, 13%-95%, 36.9%-51.9% and 2-94.7%’. line 43

              Response:

  • we corrected to ‘9-64.5%, 13-95%, 36.9-51.9% and 2-94.7%’
  1. please use the format of insects [1-4] for [1,2,3,4]
  • Response:
  • We formatted all citations throughout the text according the insects format
  1. Very big sentence, it is complicated. Please break it into two sentences. line59-62

             Response:

  • we changes and break the sentence in to two;

‘The losses that occur at these various stages, including on the field, storage, processing, and marketing in Africa, are frequently estimated to be between 20-40% [1,5]. These losses occur through the various post-harvest activities, including harvesting, handling, storing, processing, packaging, transporting, and marketing.

  1. Please provide references; ‘Insects directly destroy grain by feeding on kernels and/or indirectly by contaminating the grain with its waste, webbing and parts of the body’. line 73-74

            Response:  

  • we provide reference [4]
  1. Please alter the objectives into a text, not bullet points. line 85-87

          Response:

  • we rewrite the objectives in text as follows:

Understanding post-harvest insect pests of important crops and focusing on intervention strategies to reduce the damage are the main goals of this effort. In order to achieve this goal, it is essential that we study the different types of post-harvest insect pests that affect Ethiopian major crops, as well as the effectiveness and sufficiency of the available post-harvest management solutions across east Africa, with a focus on Ethiopia

  1. please correctly the name throughout the text, make italics: line 91, 92, 114, 146, 176, 371. 375, 378, 379

            Response:

  • We corrected all wrongly written insect names throughout the text and we put all italic names in italic font and

  1. The text is very big in the table. also this table is very complicated, percentages (%) are missing, some names are not in italics. Please fix the format: line 131

          Response:

  • The table has been modified and rectified; the heading was clarified, the name was typeset in italics, and the percentage was added.
  1. Please provide the correct table order. table 2 is under table 5: line 269

            Response:   

  • we put all table in numerical order ( Table 1, Table 5, Table 2, Table 3, and Table 4) is arranged in order (Table 1, Table 2, Table 3, Table 4, and Table 5)

  1. Other specific points
  • The entire content has been organized and checked for grammatical, spelling, and punctuation problems, including those involving commas, parenthesis, full stops, and other symbols.
  • We organized all citations and references using the structure that insects suggested.
  • The "Track changes" function indicates when the manuscript has been revised.

Round 2

Reviewer 2 Report

Dear Authors,

The text is very improved. There are a few points marked on the attacked pdf file.

Author Response

We appreciate again the supportive comments and suggestions made by the reviewer. The manuscript was revised as a consequence of taking the reviewer' comments.

  • We have revised our manuscript according to the referees’ comments
  • We checked that all references are relevant to the contents of our
  • The "Track changes" function indicates when the manuscript has been revised.

Sincerely,

Muez Berhe
